# OpenReview forum: "Safe and Efficient In-Context Learning via Risk Control"
_ICLR.cc/2026/Conference — Submitted to ICLR 2026_

### Official Review · Reviewer_FF3D · 2025-10-31

**Soundness:** 2
**Presentation:** 3
**Contribution:** 2
**Rating:** 4
**Confidence:** 4

**Summary:**

This paper proposes a risk-controlled early-exit framework for making in-context learning (ICL) safer and more efficient. The core idea is to use the model's zero-shot performance as a safe baseline and employ Distribution-Free Risk Control (DFRC) to prevent the model's performance from degrading below this baseline when exposed to potentially harmful in-context demonstrations. A key technical contribution is a risk transformation method to handle the non-standard, often negative-valued ICL loss within the Learn-then-Test framework. Experiments across multiple models and datasets aim to show that the approach controls risk as theoretically guaranteed, provides computational efficiency gains, and mitigates the impact of incorrect demonstrations.

**Strengths:**

- Novel Problem Formulation: Framing ICL safety as a risk control problem relative to a zero-shot baseline is a clear and intuitive formulation.

- Technical Adaptation: The proposed risk transformation method to handle the full range of the ICL loss within LTT is a solid technical contribution that could be useful for other applications.

- Comprehensive Evaluation: The experimental evaluation is thorough, covering multiple models, datasets, and proportions of correct/incorrect demonstrations, which robustly demonstrates the average-case behavior of the framework.

**Weaknesses:**

- Lack of Conditional Safety Guarantees: This is the most significant weakness. A safety mechanism that cannot provably protect against the specific harmful inputs it is designed to detect has limited practical utility. The paper explicitly acknowledges this shortcoming.

- Weakened Utility: The primary safety mechanism often involves reverting to zero-shot performance. In many applications, this negates the purpose of using ICL, which is to adapt and improve performance with context. The trade-off is shown, but the paper does not convincingly argue that its method provides a superior trade-off compared to simpler alternatives (e.g., using a separate classifier to detect harmful demonstrations).

- Assumptions on Adversarial Scenarios: The method is evaluated on randomly permuted labels as "incorrect demonstrations," which may not reflect the sophistication of real-world adversarial attacks or persistent user errors.

**Questions:**

Please refer to Weakness.

---

> ### Author Response · Authors · 2025-11-18
>
> We sincerely appreciate your thoughtful feedback that will help us improve our work. We also thank you for highlighting positive aspects of our work: the novelty of our problem formulation (”framing ICL safety as a risk control problem relative to a zero-shot baseline is a clear and intuitive formulation”) and significance of our technical contributions (”the proposed risk transformation…is a solid technical contribution that could be useful for other applications”), as well as the thoroughness of our evaluations (”evaluation is thorough, covering multiple models, datasets, and proportions of correct/incorrect demonstrations”).
>
> Below we address your comments:
>
> > Lack of Conditional Safety Guarantees: This is the most significant weakness. A safety mechanism that cannot provably protect against the specific harmful inputs it is designed to detect has limited practical utility. The paper explicitly acknowledges this shortcoming.
> >
>
> We appreciate your comment and would like to provide additional clarification on this limitation. We would like to point out that achieving these theoretical conditional statistical guarantees is impossible without making very strong assumptions on the underlying data distributions [1]. Note that this is not a limitation specific to our work, but it is shared by any statistical approach that relies on a finite sized calibration set (such as risk control [2,3] and conformal prediction [4]). To the best of our knowledge, there exists no safety mechanism that can ***provably*** protect against ***any*** specific harmful inputs. If you have some specific literature in mind that you would like us to address, we would greatly appreciate if you could share it with us.
>
> We would like to note that as shown in section 4.3 in the paper, when there exists some $\hat{\lambda}$ that both controls overthinking risk and preserves accuracy gains from correct demonstrations, our approach is able to find it; however, our approach cannot guarantee the existence of such a $\hat{\lambda}$.
>
> > Weakened Utility: The primary safety mechanism often involves reverting to zero-shot performance. In many applications, this negates the purpose of using ICL, which is to adapt and improve performance with context. The trade-off is shown, but the paper does not convincingly argue that its method provides a superior trade-off compared to simpler alternatives (e.g., using a separate classifier to detect harmful demonstrations).
> >
>
> We appreciate the suggestion to use a separate classifier as a baseline to detect harmful demonstrations. However, we would like to note that these approaches are complementary; our risk control approach only requires a model with any thresholded score (*including the output of such a classifier*) and a calibration set*.* However, from this perspective, using a separate classifier is a more complex approach than simply thresholding on the model itself, as it requires creating, training, and running another model. This would be a reasonable approach if the original LLM’s confidence score did not track performance well; however, we already get good results using confidence alone, so we do not see a motivation to use a separate classifier.

---

> ### Author Response · Authors · 2025-11-18
>
> > Assumptions on Adversarial Scenarios: The method is evaluated on randomly permuted labels as "incorrect demonstrations," which may not reflect the sophistication of real-world adversarial attacks or persistent user errors.
> >
>
> **TL;DR we focus specifically on non-obvious tampering with ICL demonstrations; our method is a complementary safeguard that dynamically limits performance degradation without requiring model retraining**
>
> Thank you for the helpful feedback. We agree that our framing of risks could be clearer. The scenario we describe in the intro is a very specific attack scenario: one may tamper with in-context learning (ICL) demonstrations in a way that may not be obvious to a model maintainer. Given that ICL is a broad and general-purpose capability and used in the design of various systems, we expect this to be an application of increasing importance. Exactly how the adversary may be able to access or modify the ICL demonstrations is something that we have intentionally abstracted away to decouple the problem and focus our work on an isolated layer of the problem.
>
> Existing adversarial and jailbreak defense techniques often require some model re-training [5,6]; in contrast, our method provides distribution-free safety guarantees without assuming any prior knowledge of specific error types. We do not view these two approaches as direct competitors; rather, they are complementary approaches to LLM safety. Our framework provides a built-in safeguard by dynamically limiting performance degradation relative to a safe zero-shot baseline.
>
> We appreciate the comment and will revise the paper to explicitly clarify this scope and add discussion distinguishing our approach from adversarial defenses, emphasizing that our method complements but does not replace specialized techniques for explicit attack settings.
>
> **We look forward to addressing any remaining questions during the rebuttal and hope these updates warrant a higher evaluation.**
>
> ### References
>
> **[1]** Rina Foygel Barber, Emmanuel J. Candès, Aaditya Ramdas, Ryan J. Tibshirani. *The limits of distribution-free conditional predictive inference.* Information and Inference 2019.
>
> **[2]** Anastasios N. Angelopoulos, Stephen Bates, Emmanuel J. Candès, Michael I. Jordan, Lihua Lei. *Learn then Test: Calibrating Predictive Algorithms to Achieve Risk Control.* Annals of Applied Statistics 2025.
>
> **[3]** Metod Jazbec, Alexander Timans, Tin Hadži Veljković, Kaspar Sakmann, Dan Zhang, Christian A. Naesseth, Eric Nalisnick. *Fast yet Safe: Early-Exiting with Risk Control.* NeurIPS 2024.
>
> **[4]** Tiffany Ding, Anastasios Angelopoulos, Stephen Bates, Michael I. Jordan, Ryan J Tibshirani. *Class-Conditional Conformal Prediction with Many Classes.* NeurIPS 2023.
>
> **[5]** Minseon Kim, Jin Myung Kwak, Lama Alssum, Bernard Ghanem, Philip Torr, David Krueger, Fazl Barez, Adel Bibi. *Rethinking Safety in LLM Fine-tuning: An Optimization Perspective.* COLM 2025.
>
> **[6]** Josef Dai, Xuehai Pan, Ruiyang Sun, Jiaming Ji, Xinbo Xu, Mickel Liu, Yizhou Wang, Yaodong Yang. *Safe RLHF: Safe Reinforcement Learning from Human Feedback.* ICLR 2023.

---

### Official Review · Reviewer_XUPs · 2025-10-31

**Soundness:** 3
**Presentation:** 4
**Contribution:** 2
**Rating:** 4
**Confidence:** 3

**Summary:**

The paper tackles the problem of LLM-based classification given in-context demonstrations, in a setting where the demonstrations might be helpful but could also be incorrect/adversarial.

The proposed method computes confidence scores for early-exit classifications after each layer and makes a classification as soon as a certain confidence threshold is reached. If by the final layer, the confidence threshold hasn't been reached, the method falls back to a baseline of classifying without using any of the in-context examples.

The paper shows that with an appropriate confidence threshold, this method can often benefit from correct demonstrations (i.e. outperform the zero-shot baseline) without being harmed much by incorrect demonstrations, on tasks like sentiment classification or hate speech detection.

**Strengths:**

- It's an interesting approach to the problem of overthinking on incorrect examples, which is still able to benefit from correct examples
- The paper is very well written and describes experiments etc. clearly and in enough detail
- There are a good amount of ablations, and more broadly the experiment design seems very thoughtful
- The method seems to work well in many of the studied settings

Overall, I think this is a nice idea that's executed well.

**Weaknesses:**

- The paper discusses adversarial demonstrations as one important motivation in the abstract/introduction. But I think the methods and results don't imply much adversarial robustness and seem more centrally about examples that are simply incorrect (e.g. the experiments simply flip the labels of examples). For many adversarial threat models, an attacker would also be able to modify the text of the examples themselves (and/or could in principle try to modify labels more adversarially than making all of them incorrect: maybe a mix of correct and incorrect labels could lead to inflated confidence scores and still give incorrect early-exit answers?) I think the paper would be more accurately presented if it didn't focus its motivation on the adversarial case (or at least made it very clear that it does not provide assurances against actual adversaries if I understand correctly). (This is the only reason I rated Soundness as 3 instead of 4.)
- It's unclear whether the method could be generalized well to the kinds of broader tasks that the introduction mentions (such as a coding agent). If it only works for classification using few-shot prompted LLMs, where the few-shot examples can't be trusted, that is a rather narrow application.
- In cases where the model developer/deployer is providing the few-shot demonstrations, it might be better to verify the demonstrations and ensure they are correct rather than implement this method. So I think the main potential of the method comes from cases where few-shot examples are provided dynamically, e.g. if a model is used for many different user-defined classification tasks. But then fig. 3 makes me worried because it suggests that the right lambda will depend a lot on the specific classification task, which creates more effort for each new classification task (and again raises the question whether trying to find correct examples might be better). Similar to the previous bullet, this doesn't invalidate the method, but I think it meaningfully restricts its applicability.

I'm open to updating my score if these issues are addressed (whether through changes to the presentation, additional experiments, or convincing examples showing that I'm wrong about the limited applicability.)

**Questions:**

1. In fig. 3, what exactly does "accuracy relative to zero-shot" mean? Would 0.2 mean that the accuracy is 20 percentage points higher than the zero-shot baseline? Then it seems even incorrect demonstrations outperform the zero-shot baseline, since the y-axis doesn't have negative numbers. Or is there some re-calibration to make the zero-shot baseline have 50% accuracy? But then on AG News, even correct demonstrations would be worse than the baseline.
2. Re applicability of the method: do you have specific practical applications in mind where you think the method in its current form would actually be the best choice? Alternatively, what kinds of future extensions do you think could be feasible that would have broader applicability?

---

> ### Author Response · Authors · 2025-11-18
>
> We sincerely appreciate your thoughtful feedback that will help us improve our work. We also thank you for highlighting positive aspects of our work: we effectively balance helpful and harmful demonstrations (“this method can often benefit from correct demonstrations without being harmed much by incorrect demonstrations”), and we effectively demonstrate the effectiveness of our approach (”[works] well in many of the studied settings”) through very well-designed settings (”the experiment design seems very thoughtful”).
>
> Below we address your comments:
>
> > The paper discusses adversarial demonstrations as one important motivation in the abstract/introduction. But I think the methods and results don't imply much adversarial robustness and seem more centrally about examples that are simply incorrect (e.g. the experiments simply flip the labels of examples). For many adversarial threat models, an attacker would also be able to modify the text of the examples themselves (and/or could in principle try to modify labels more adversarially than making all of them incorrect: maybe a mix of correct and incorrect labels could lead to inflated confidence scores and still give incorrect early-exit answers?) I think the paper would be more accurately presented if it didn't focus its motivation on the adversarial case (or at least made it very clear that it does not provide assurances against actual adversaries if I understand correctly). (This is the only reason I rated Soundness as 3 instead of 4.)
> >
>
> **TL;DR we focus specifically on non-obvious tampering with ICL demonstrations; our method is a complementary safeguard that dynamically limits performance degradation without requiring model retraining**
>
> Thank you for the helpful feedback. We agree that our framing of “security risks” could be clearer. The scenario we describe in the intro is a very specific attack scenario: one may tamper with in-context learning (ICL) demonstrations in a way that may not be obvious to a model maintainer. Given that ICL is a broad and general-purpose capability and used in the design of various systems, we expect this to be an application of increasing importance. Exactly how the adversary may be able to access or modify the ICL demonstrations is something that we have intentionally abstracted away to decouple the problem and focus our work on an isolated layer of the problem.
>
> Existing adversarial and jailbreak defense techniques often require some model re-training [5,6]; in contrast, our method provides distribution-free safety guarantees without assuming any prior knowledge of specific error types. We do not view these two approaches as direct competitors; rather, they are complementary approaches to LLM safety. Our framework provides a built-in safeguard by dynamically limiting performance degradation relative to a safe zero-shot baseline.
>
> We appreciate the comment and will revise the paper to explicitly clarify this scope and add discussion distinguishing our approach from adversarial defenses, emphasizing that our method complements but does not replace specialized techniques for explicit attack settings.
>
> > It's unclear whether the method could be generalized well to the kinds of broader tasks that the introduction mentions (such as a coding agent). If it only works for classification using few-shot prompted LLMs, where the few-shot examples can't be trusted, that is a rather narrow application.
> >
>
> **TL;DR our approach is designed to generalize broadly across task types, only requiring inputs which are agnostic to task format; we are exploring extension to open-ended generation tasks**
>
> We appreciate this comment. While our experiments focus on classification-style tasks, the proposed framework is designed to generalize broadly across task types because it operates on the model’s layerwise confidence and a measure of relative risk: concepts that are agnostic to task format. The same mechanism that detects “overthinking” and limits degradation relative to a safe zero-shot baseline can naturally extend to open-ended generative settings where unsafe or low-quality examples cause the model to deviate from safe or intended behavior. Existing work has studied model confidence and token entropy as a means to improve open-ended reasoning [1, 2] and avoid unsafe responses on open-ended QA tasks [3]. We can apply similar ideas for a straightforward extension of our work to more general open-ended LLM tasks, using an approach such as [4].

---

> ### Author Response · Authors · 2025-11-18
>
> > In cases where the model developer/deployer is providing the few-shot demonstrations, it might be better to verify the demonstrations and ensure they are correct rather than implement this method. So I think the main potential of the method comes from cases where few-shot examples are provided dynamically, e.g. if a model is used for many different user-defined classification tasks. But then fig. 3 makes me worried because it suggests that the right lambda will depend a lot on the specific classification task, which creates more effort for each new classification task (and again raises the question whether trying to find correct examples might be better). This meaningfully restricts applicability.
> >
>
> We appreciate the comment pointing out the potentially limited applicability of our work. We disagree that demonstrations should be verified manually for correctness instead of implementing our method; when a model has been deployed, model developers have no control over the demonstrations provided by system users, and manually verifying all demonstrations for a large-scale system with many users is infeasible. Regarding the lambda-choice being specific to a particular classification task: on a practical deployment, we can also choose such a lambda that works across an aggregation of multiple tasks. However, finding a lambda for each type of task can be performed post-hoc and does not require any model re-training, so it is fairly efficient and easy to find a lambda for each specific task.
>
> > In fig. 3, what exactly does "accuracy relative to zero-shot" mean? Would 0.2 mean that the accuracy is 20 percentage points higher than the zero-shot baseline? Then it seems even incorrect demonstrations outperform the zero-shot baseline, since the y-axis doesn't have negative numbers. Or is there some re-calibration to make the zero-shot baseline have 50% accuracy? But then on AG News, even correct demonstrations would be worse than the baseline.
> >
>
> We appreciate the point here and apologize for the confusion. This was an error on our part; the y-axis should read “accuracy” (not relative to zero-shot). We have fixed the axis label and updated the plot in the paper.
>
> > Re applicability of the method: do you have specific practical applications in mind where you think the method in its current form would actually be the best choice? Alternatively, what kinds of future extensions do you think could be feasible that would have broader applicability?
> >
>
> **TL;DR our method is useful for any user-facing LLM system (e.g., coding assistants, co-pilots) using ICL for customization, to limit risk from ICL demos; with future extensions to safety on multi-turn generative tasks**
>
> In its current form, our method is best suited for user-facing LLM systems that rely on in-context learning for customization without fine-tuning, such as coding assistants, data-analysis copilots, or domain-specific chat systems. In these settings, users often provide task examples of varying quality, and our approach offers a lightweight, built-in safeguard that limits degradation from poor or unsafe demonstrations while enabling efficiency gains from helpful ones. For instance, a coding assistant could automatically revert to its zero-shot behavior when a user-supplied example introduces insecure code, or a customer-support bot could disregard inconsistent few-shot examples that reduce factual accuracy.
>
> Looking forward, we see several feasible extensions for broader applicability. Future work could integrate our risk-controlled early-exit mechanism into multi-turn generative tasks (e.g., summarization, reasoning, or dialogue). As mentioned in a response to your earlier comment, existing work has studied model confidence and token entropy as a means to improve open-ended reasoning [1, 2] and avoid unsafe responses on open-ended QA tasks [3]. We can apply similar ideas for a straightforward extension of our work to more general LLM tasks. These directions would extend the same core principle - maintaining bounded risk relative to a trusted baseline - to a wider range of LLM-based applications.
>
> **We look forward to addressing any remaining questions during the rebuttal and hope these updates warrant a higher evaluation.**

---

> > ### Comment · Reviewer_XUPs · 2025-11-24
> >
> > Thank you for the detailed reply!
> >
> > On the security point, I appreciate the plans to revise the paper. But I'm still confused by some of the phrasing, which makes me unsure whether we are on the same page about the extent to which the method is well-described as an adversarial defence. My concern is not primarily whether the proposed method can or cannot be combined with jailbreak defenses like finetuning. What I'd like the paper to make clear is to what extent the method by itself should be viewed as a defense against adversarial ICL demonstrations.
> >
> > > The scenario we describe in the intro is a very specific attack scenario: one may tamper with in-context learning (ICL) demonstrations in a way that may not be obvious to a model maintainer
> >
> > The attack scenario that the method provides guarantees against to me seems much more specific than general attacks that tampers with ICL demos in subtle ways. As I mentioned, the paper focuses on what happens if the labels of in-context examples are all wrong. An attacker who tampers with ICL examples could also modify the examples themselves.
> >
> > The response mentions "safety guarantees" and a "safeguard by dynamically limiting performance degradation", and I think it will be important to make clear that "safety" here only refers to safety against incorrectly labeled demonstrations, not safety against adversarially chosen demonstrations. Maybe the method also provides safety against more general adversaries, but since this isn't evaluated, I would recommend making this distinction very clear and de-emphasizing the adversarial framing overall. (Alternatively, it would of course be interesting to actually study the adversarial robustness of the method.)
> >
> > On applicability, I still think that the current results don't demonstrate that the method would be useful in practically useful situations, e.g. I think the coding assistant example is much more ambitious than what the paper currently demonstrates. Of course the paper is still a useful step, but for now I largely stand by the limitations I originally raised. I look forward to seeing results in a generation setting as mentioned in the top-level reply.
> >
> > > our method is useful for any user-facing LLM system (e.g., coding assistants, co-pilots) using ICL for customization
> >
> > I think the key question is how well-supported this claim is by the results in the paper. I appreciate the pointers for how the method could be applied to non-classification tasks. But based on current results, I'm overall skeptical that this would work with only simple extensions of the current method, and so I still think the applicability is a big question mark without actual experiments in more realistic settings like this.
> >
> > To flesh this out, applying the method to e.g. a coding assistant would (I think) look something like this:
> > - Use entropy over the next token (or some other metric) to decide on early exit.
> > - Do this for ever generated token I assume?
> > - Pick a single lambda that works well across all the different types of queries that users might send to the assistant.
> >
> > There are a huge number of uncertainties here. E.g. how well would using some metric over next tokens work, which seems meaningfully different from binary classification? Does applying the method to generating many tokens work, or does generation quality quickly degrade (e.g. as the generation becomes off-policy)? And given that the experiments in the paper pick different thresholds across different datasets that are all binary classification tasks, picking a single threshold for a much larger variety of queries, and in a generation setting, seems difficult.
> >
> > > Regarding the lambda-choice being specific to a particular classification task: on a practical deployment, we can also choose such a lambda that works across an aggregation of multiple tasks
> >
> > Just to clarify: that there would be a single lambda that would work well across an aggregation of multiple tasks is a guess rather than an empirical fact, right? It seems easily possible that for a realistic range of tasks, no single lambda would provide good performance from what I can tell. (And the fact that the optimal lambda differs significantly between different experiments in the paper is what makes me especially worried about this.)
> >
> > (I agree that of course manual verification doesn't scale with user-provided examples! My review only suggested manual as an option for cases where the examples *don't* come from users. So my only point there was that a broad range of tasks, with user-specified examples, is the main case where this paper's method seems necessary, so a broad task range seems like the key setting, and thus the question of lambda choice is important. Sorry if this caused confusion, I'm happy to drop the manual verification point if we're in agreement anyway that user-provided examples in tasks that aren't too narrow are the interesting regime.)

---

> > > ### Author Response · Authors · 2025-11-28
> > >
> > > Thank you for your continued engagement with our rebuttal! We respond to your concerns below.
> > >
> > > > On the security point, I appreciate the plans to revise the paper. But I'm still confused by some of the phrasing, which makes me unsure whether we are on the same page about the extent to which the method is well-described as an adversarial defence…The attack scenario that the method provides guarantees against to me seems much more specific than general attacks that tampers with ICL demos in subtle ways. An attacker who tampers with ICL examples could also modify the examples themselves.
> > > >
> > >
> > > We appreciate your concern regarding the scope of our method's safety guarantees and the adversarial framing. We do mention an adversarial setting as building motivation for our general approach, but would like to clarify that we do not make a direct claim of contribution in an adversarial setting. We appreciate your feedback and will further de-emphasize this setting in the abstract and introduction.
> > >
> > > > I still think that the current results don't demonstrate that the method would be useful in practically useful situations, e.g. I think the coding assistant example is much more ambitious than what the paper currently demonstrates. Of course the paper is still a useful step, but for now I largely stand by the limitations I originally raised. I look forward to seeing results in a generation setting as mentioned in the top-level reply.
> > > >
> > >
> > > We appreciate the acknowledgement that our work is a useful step and are continuing to work on open-ended generation tasks to address the limitations you mentioned, building on approaches such as [1] and [2]. We will post an update soon!
> > >
> > > > I think the key question is how well-supported this claim [that our work is useful for any user-facing LLM system] is by the results in the paper. I appreciate the pointers for how the method could be applied to non-classification tasks. But based on current results, I'm overall skeptical that this would work with only simple extensions of the current method. E.g. how well would using some metric over next tokens work, which seems meaningfully different from binary classification? Does applying the method to generating many tokens work, or does generation quality quickly degrade (e.g. as the generation becomes off-policy)?
> > > >
> > >
> > > As mentioned in our response above, work such as [1] and [2] makes this a straightforward extension, as they have already developed approaches for early-exit LLMs with open-ended generation and confidence thresholding. We can extend their work by adding in-context demonstrations and using our risk control approach, which we are currently doing via new experiments.
> > >
> > > > Just to clarify: that there would be a single lambda that would work well across an aggregation of multiple tasks is a guess rather than an empirical fact, right? It seems easily possible that for a realistic range of tasks, no single lambda would provide good performance from what I can tell. (And the fact that the optimal lambda differs significantly between different experiments in the paper is what makes me especially worried about this.)
> > > >
> > >
> > > We reproduced Figure 3 from the paper, but aggregated across all our datasets. We find that there still exists a good lambda that loses no more than 5% of the accuracy gains from correct demonstrations while still doing better than the full model given incorrect demonstrations. In general, increasing the granularity of the lambdas (i.e. the number of lambda-values evaluated) can further increase the number of such lambdas. Again, our approach cannot guarantee the existence of such a lambda, but it is quite a reasonable assumption that one may exist, even when aggregating across many tasks. We will mention these results in the paper.
> > >
> > > > I'm happy to drop the manual verification point if we're in agreement anyway that user-provided examples in tasks that aren't too narrow are the interesting regime.
> > > >
> > >
> > > We appreciate your comment and agree with the point that user-provided examples in tasks that aren’t too narrow are the setting of interest.
> > >
> > > **[1]** Tal Schuster, Adam Fisch, Jai Gupta, Mostafa Dehghani, Dara Bahri, Vinh Q. Tran, Yi Tay, Donald Metzler. *Confident Adaptive Language Modeling.* NeurIPS 2022.
> > >
> > > **[2]** Metod Jazbec, Alexander Timans, Tin Hadži Veljković, Kaspar Sakmann, Dan Zhang, Christian A. Naesseth, Eric Nalisnick. *Fast yet Safe: Early-Exiting with Risk Control.* NeurIPS 2024.

---

> ### Author Response · Authors · 2025-11-18
>
> ### References
>
> **[1]** Xi Wang, James McInerny, Lequn Wang, Nathan Kallus. *Entropy After </Think> for Reasoning Model Early-Exiting*. arXiv 2025.
>
> **[2]** Yichao Fu, Xuewei Wang, Yuandong Tian, Jiawei Zhao. *Deep Think with Confidence.* arXiv 2025.
>
> **[3]** William Jurayj, Jeffery Cheng, Benjamin Van Durme. *Is That Your Final Answer? Test-Time Scaling Improves Selective Question Answering.* ACL 2025.
>
> **[4]** Tal Schuster, Adam Fisch, Jai Gupta, Mostafa Dehghani, Dara Bahri, Vinh Q. Tran, Yi Tay, Donald Metzler. *Confident Adaptive Language Modeling.* NeurIPS 2022.
>
> **[5]** Minseon Kim, Jin Myung Kwak, Lama Alssum, Bernard Ghanem, Philip Torr, David Krueger, Fazl Barez, Adel Bibi. *Rethinking Safety in LLM Fine-tuning: An Optimization Perspective.* COLM 2025.
>
> **[6]** Josef Dai, Xuehai Pan, Ruiyang Sun, Jiaming Ji, Xinbo Xu, Mickel Liu, Yizhou Wang, Yaodong Yang. *Safe RLHF: Safe Reinforcement Learning from Human Feedback.* ICLR 2023.

---

### Official Review · Reviewer_hBNu · 2025-11-01

**Soundness:** 3
**Presentation:** 3
**Contribution:** 3
**Rating:** 4
**Confidence:** 4

**Summary:**

This paper examines the potential for overthinking, as well as the safety and efficiency issues raised by LLMs in ICL. The authors propose a safe and efficient early-exit ICL framework by defining the ICL risk with respect to a zero-sample baseline and utilizing a distribution-independent hierarchical risk control method to dynamically decide when to early-exit in the reasoning process, thereby avoiding unnecessary computation. Experiments demonstrate the effectiveness of the method on eight classification tasks.

**Strengths:**

1. Sufficient theoretical support. The paper provides rigorous theoretical proofs in the appendix to formalize the effectiveness of risk transformation strategies in controlling ICL risk.
2. Novel framework. The authors introduce a risk control framework to the LLM security problem, where overthinking is weighed against performance loss through dynamic threshold selection.

**Weaknesses:**

1. Lack of clarity on the types of security problems. Although the paper claims to control “security risks,” it is not clear what types of security problems (e.g., demonstration poisoning, jailbreak attacks, fake content generation, etc.) are targeted. Different types of security risks correspond to different attacks and defense methods, and the authors should clarify the scope of application and attack assumptions. In addition, several of the above attacks have explicit methods, and authors should discuss these methods.
2. Insufficient articulation of methodological innovations with existing frameworks. The authors extend based on existing risk control theories but do not clearly articulate what new mechanisms or assumptions are added to this foundation and how these extensions specifically address the security risks of LLMs.
3. lack of comparison with existing defense approaches. The related work section does not explain how existing security mitigation techniques (e.g., adversarial training, risk weighting, jailbreak detection) compare to the risk control framework, and the experiments do not provide comparative validation, so the current results are insufficient to fully demonstrate the effectiveness of the approach.
4. Efficiency advantages are not quantified. Although the title emphasizes EFFICIENCY, the text does not provide algorithm complexity analysis, reasoning latency or computational overhead comparison, resulting in EFFICIENCY lackingempirical support.
5. Single task type. The experiments are all categorization tasks (e.g., SST2, AG News, TweetEval, etc.) and do not cover generative/open-ended tasks, and the generality of the method for LLM remains to be verified.

**Questions:**

Please examine the weaknesses.

---

> ### Author Response · Authors · 2025-11-18
>
> We sincerely appreciate your thoughtful feedback that will help us improve our work. We also thank you for highlighting positive aspects of our work: a rigorous, novel framework for enhancing security in LLMs (”the authors introduce a risk control framework to the LLM security problem [with] sufficient theoretical support”) while simultaneously achieving computational efficiency (”dynamically [deciding] when to early-exit in the reasoning process, thereby avoiding unnecessary computation”). We would also like to refer the reviewer to our general comment, in which we reiterate the main contributions of our work.
>
> Below we address your comments:
>
> > Lack of clarity on the types of security problems. Although the paper claims to control “security risks,” it is not clear what types of security problems (e.g., demonstration poisoning, jailbreak attacks, fake content generation, etc.) are targeted. Different types of security risks correspond to different attacks and defense methods, and the authors should clarify the scope of application and attack assumptions. In addition, several of the above attacks have explicit methods, and authors should discuss these methods.
> >
>
> > Lack of comparison with existing defense approaches. The related work section does not explain how existing security mitigation techniques (e.g., adversarial training, risk weighting, jailbreak detection) compare to the risk control framework, and the experiments do not provide comparative validation, so the current results are insufficient to fully demonstrate the effectiveness of the approach.
> >
>
> **TL;DR we focus specifically on non-obvious tampering with ICL demonstrations; our method is a complementary safeguard that dynamically limits performance degradation without requiring model retraining**
>
> Thank you for the helpful feedback. We agree that our framing of “security risks” could be clearer. The scenario we describe in the intro is a very specific attack scenario: one may tamper with in-context learning (ICL) demonstrations in a way that may not be obvious to a model maintainer. Given that ICL is a broad and general-purpose capability and used in the design of various systems, we expect this to be an application of increasing importance. Exactly how the adversary may be able to access or modify the ICL demonstrations is something that we have intentionally abstracted away to decouple the problem and focus our work on an isolated layer of the problem.
>
> Existing adversarial and jailbreak defense techniques often require some model re-training [1,2]; in contrast, our method provides distribution-free safety guarantees without assuming any prior knowledge of specific error types. We do not view these two approaches as direct competitors; rather, they are complementary approaches to LLM safety. Our framework provides a built-in safeguard by dynamically limiting performance degradation relative to a safe zero-shot baseline.
>
> We appreciate the comment and will revise the paper to explicitly clarify this scope and add discussion distinguishing our approach from existing adversarial defenses, emphasizing that our method complements but does not replace specialized techniques for explicit attack settings.

---

> ### Author Response · Authors · 2025-11-18
>
> > Insufficient articulation of methodological innovations with existing frameworks. The authors extend based on existing risk control theories but do not clearly articulate what new mechanisms or assumptions are added to this foundation and how these extensions specifically address the security risks of LLMs.
> >
>
> **TL;DR prior work on early-exit and risk control were developed independently and for unrelated goals; we are the first to integrate them, and we are also the first to apply risk control for harmful demos in ICL**
>
> Existing work on risk control has not, to the best of our knowledge, been applied to an in-context learning setting; we provide the first rigorous safety guarantees that we are aware of in this setting. Prior early-exit and risk-control methods were developed independently and for unrelated goals [2,3], whereas our work repurposes and integrates them to tackle the novel safety challenge of harmful demonstrations in ICL. In comparison to existing risk control methods, not only are we the first to use risk control for in-context learning (which requires a new ICL loss that reflects overthinking), but we also propose the technical extensions necessary to make risk control practical in the scenario of mixed-quality demonstrations (i.e. our loss-scaling approach). The novelty of our approach in these respects has also been acknowledged by reviewer FF3D.
>
> Among the literature on early-exit models, our work is the first to apply early-exit for in-context learning. The closest paper to our work is by Halawi et al [1], but they statically prune the last layers, whereas we perform dynamic early-exiting via confidence thresholding, enabling more granular risk control. **We are currently running additional experiments to directly compare our approach with static early-exit to demonstrate the comparative effectiveness of our approach.** We believe this is a significant and important contribution as our work makes a bridge between two active areas of research that can benefit each other.
>
> We appreciate the comment and will make the novelty and significance of our contributions more clear in the paper.
>
> > Efficiency advantages are not quantified. Although the title emphasizes EFFICIENCY, the text does not provide algorithm complexity analysis, reasoning latency or computational overhead comparison, resulting in EFFICIENCY lacking empirical support.
> >
>
> We would like to point out that we do report strong empirical results on efficiency gains across all tasks and models. Please refer to Fig. 6 in the paper, where we show that we achieve greater efficiency gains than an alternative risk control approach. We quantify our efficiency gains as the average number of layers of the model which are skipped at test-time using our approach, which is directly proportional to the reduction in computation needed to make a prediction (i.e. FLOPs). This is a standard measure of efficiency used in early-exit literature (e.g. [6]). We define this measure of efficiency gains in Section 4.2 where we report results; however, we will provide this definition earlier in the paper as well for clarity.
>
> > Single task type. The experiments are all categorization tasks (e.g., SST2, AG News, TweetEval, etc.) and do not cover generative/open-ended tasks, and the generality of the method for LLM remains to be verified.
> >
>
> **TL;DR our approach is designed to generalize broadly across task types, only requiring inputs which are agnostic to task format; we are exploring extension to open-ended generation tasks**
>
> We appreciate this comment. While our experiments focus on classification-style tasks, the proposed framework is designed to generalize broadly across task types because it operates on the model’s layerwise confidence and a measure of relative risk: concepts that are agnostic to task format. The same mechanism that detects “overthinking” and limits degradation relative to a safe zero-shot baseline can naturally extend to open-ended generative settings where unsafe or low-quality examples cause the model to deviate from safe or intended behavior. Existing work has studied model confidence and token entropy as a means to improve open-ended reasoning [3,4] and avoid unsafe responses on open-ended QA tasks [5].
>
> We can apply similar ideas for a straightforward extension of our work to more general, open-ended LLM tasks. In particular, as we call out in our general comment, we are looking into an open-ended generation task such as [7] using our early-exit approach as part of the rebuttal. These will be available soon, and we will post an update with results during the rebuttal period.
>
> **We look forward to addressing any remaining questions during the rebuttal and hope these updates warrant a higher evaluation.**

---

> ### Author Response · Authors · 2025-11-18
>
> ### References
>
> **[1]** Minseon Kim, Jin Myung Kwak, Lama Alssum, Bernard Ghanem, Philip Torr, David Krueger, Fazl Barez, Adel Bibi. *Rethinking Safety in LLM Fine-tuning: An Optimization Perspective.* COLM 2025.
>
> **[2]** Josef Dai, Xuehai Pan, Ruiyang Sun, Jiaming Ji, Xinbo Xu, Mickel Liu, Yizhou Wang, Yaodong Yang. *Safe RLHF: Safe Reinforcement Learning from Human Feedback.* ICLR 2023.
>
> **[3]** Xi Wang, James McInerny, Lequn Wang, Nathan Kallus. *Entropy After </Think> for Reasoning Model Early-Exiting*. arXiv 2025.
>
> **[4]** Yichao Fu, Xuewei Wang, Yuandong Tian, Jiawei Zhao. *Deep Think with Confidence.* arXiv 2025.
>
> **[5]** William Jurayj, Jeffery Cheng, Benjamin Van Durme. *Is That Your Final Answer? Test-Time Scaling Improves Selective Question Answering.* ACL 2025.
>
> **[6]** Metod Jazbec, Alexander Timans, Tin Hadži Veljković, Kaspar Sakmann, Dan Zhang, Christian A. Naesseth, Eric Nalisnick. *Fast yet Safe: Early-Exiting with Risk Control.* NeurIPS 2024.
>
> **[7]** Tal Schuster, Adam Fisch, Jai Gupta, Mostafa Dehghani, Dara Bahri, Vinh Q. Tran, Yi Tay, Donald Metzler. *Confident Adaptive Language Modeling.* NeurIPS 2022.

---

### Official Review · Reviewer_dbng · 2025-11-05

**Soundness:** 3
**Presentation:** 2
**Contribution:** 2
**Rating:** 4
**Confidence:** 3

**Summary:**

This paper proposes a risk-controlled early-exit framework for safe and efficient in-context learning in large language models. The authors leverage zero-shot predictions as a safety baseline and apply distribution-free risk control (DFRC) to ensure that in-context examples do not degrade performance below this baseline. To mitigate overthinking caused by harmful demonstrations, the method employs early-exit mechanisms based on confidence thresholds. A novel ICL-specific loss is introduced to quantify the extent of overthinking, and the Learn-then-Test (LTT) framework is adapted to select thresholds under non-monotonic and potentially negative-valued losses. Experimental results across eight classification tasks and four models demonstrate that the approach effectively controls predictive risk while achieving significant computational speedups.

**Strengths:**

1. The paper proposes a clear and reasonable approach to simultaneously address safety and efficiency in in-context learning, with writing that is easy to follow.

2. The method is theoretically grounded through the use of a risk control framework, and the experimental results demonstrate its effectiveness.

**Weaknesses:**

1. The motivation for jointly addressing both safety and efficiency is unclear; it is not well justified why these two objectives must be addressed together rather than separately.

2. The method and analysis primarily adapts existing techniques, such as early exiting and risk control, which makes the overall contribution appear somewhat incremental.

3. Some of the mathematical formulations (e.g., Equations (1) and (2)) are unnecessarily verbose and could be streamlined for clarity and conciseness.

**Questions:**

None

---

> ### Author Response · Authors · 2025-11-18
>
> We sincerely appreciate your thoughtful feedback that will help us improve our work. We also thank you for highlighting positive aspects of our work: a novel effective framework addressing both risk and efficiency (”effectively controls predictive risk while achieving significant computational speedups”) and well-supported by theory (”the method is theoretically grounded through the use of a risk control framework”). We would also like to refer the reviewer to our general comment, in which we reiterate the main contributions of our work.
>
> Below we address your comments:
>
> > The motivation for jointly addressing both safety and efficiency is unclear; it is not well justified why these two objectives must be addressed together rather than separately.
> >
>
> We appreciate the question, and would like to emphasize that our motivation is **not** to jointly address safety and efficiency. Our motivation is safety (mitigating the influence of harmful in-context demonstrations and overthinking), while inference efficiency is a nice side benefit. The reason we get those efficiency gains “for free” is because we use early-exiting to prevent overthinking, and early-exiting has traditionally been used as an approach to achieve efficiency gains. So we hope that our work will lead to more research looking into repurposing early-exit models not only for efficiency, but also for safety to mitigate overthinking.
>
> We thank the reviewer for bringing this up and will be sure to make this (i.e., the fact that we are not aiming to address safety and efficiency jointly) more clear in the paper. Additionally, we plan to change the title of the paper to “Safe In-Context Learning via Risk Control and Early-Exiting”, more clearly indicating that our focus is on safety.
>
> > The method and analysis primarily adapts existing techniques, such as early exiting and risk control, which makes the overall contribution appear somewhat incremental.
> >
>
> Existing work on risk control has not, to the best of our knowledge, been applied to an in-context learning setting; we provide the first rigorous safety guarantees that we are aware of in this setting. Prior early-exit and risk-control methods were developed independently and for unrelated goals [2,3], whereas our work repurposes and integrates them to tackle the novel safety challenge of harmful demonstrations in ICL. In comparison to existing risk control methods, not only are we the first to use risk control for in-context learning (which requires a new ICL loss that reflects overthinking), but we also propose the technical extensions necessary to make risk control practical in the scenario of mixed-quality demonstrations (i.e. our loss-scaling approach). The novelty of our approach in these respects has also been acknowledged by reviewers hBNu and FF3D.
>
> Among the literature on early-exit models, our work is the first to apply early-exit for in-context learning. The closest paper to our work is by Halawi et al [1], but they statically prune the last layers, whereas we perform dynamic early-exiting via confidence thresholding, enabling more granular risk control. **We are currently running additional experiments to directly compare our approach with static early-exit to demonstrate the comparative effectiveness of our approach, as detailed in our general comment.** We believe this is a significant and important contribution as our work makes a bridge between two active areas of research that can benefit each other.
>
> We appreciate the comment and will make the novelty and significance of our contributions more clear in the paper.
>
> > Some of the mathematical formulations (e.g., Equations (1) and (2)) are unnecessarily verbose and could be streamlined for clarity and conciseness.
> >
>
> We appreciate the insightful comment. We have rewritten equation 2 in the paper as follows to improve conciseness and clarity:
>
> $$
> \bar{y}\_{\lambda} := \begin{cases}        \hat{y}\_{\lambda} & \mathfrak{C}\_i \geq \lambda, \ \exists i \in \\{1, ..., L\\} \\\\        \mathbf{\arg\max\limits\_{k \in \mathcal{Y}} p\_L\left(k | x \right)} & \textbf{otherwise.}    \end{cases}
> $$
>
> If you have further suggestions on improving clarity of Eq. 2 and Eq. 3 we would be happy to hear them!
>
> **We look forward to addressing any remaining questions during the rebuttal and hope these updates warrant a higher evaluation.**
>
> ### References
>
> **[1]** Danny Halawi, Jean-Stanislas Denain, Jacob Steinhardt. *Overthinking the Truth: Understanding How Language Models Process False Demonstrations.* ICLR 2024.
>
> **[2]** Anastasios N. Angelopoulos, Stephen Bates, Emmanuel J. Candès, Michael I. Jordan, Lihua Lei. *Learn then Test: Calibrating Predictive Algorithms to Achieve Risk Control. * Annals of Applied Statistics 2025.
>
> **[3]** Metod Jazbec, Alexander Timans, Tin Hadži Veljković, Kaspar Sakmann, Dan Zhang, Christian A. Naesseth, Eric Nalisnick. *Fast yet Safe: Early-Exiting with Risk Control.* NeurIPS 2024.

---

### Author Response · Authors · 2025-11-18

Dear Reviewers,

We thank you for your time and thoughtful feedback which will help us improve our paper. We would like to reiterate our key contributions:

- We formulate a *new safe early-exit model* that uses the zero-shot baseline as an explicit reference for safe behavior. To the best of our knowledge, this is the first work to apply dynamic early-exiting for ensuring safety.
- We propose a *novel in context learning (ICL)-specific loss* to measure “overthinking” by quantifying performance relative to the zero-shot model.
- We introduce a *domain-preserving risk transformation* that extends Learn-then-Test (LTT) risk control to losses with negative values, enabling rigorous safety guarantees while still exploiting performance and efficiency gains from helpful demonstrations.

This combination is theoretically novel, empirically validated across eight tasks and four models under multiple distributions of correct and incorrect demonstrations, and establishes for the first time formal distribution-free safety guarantees for LLMs adapting to arbitrary in-context demonstrations.

We would further like to emphasize that our focus in this work is on *safety*, and *efficiency* is merely a side benefit of our approach. For this reason, we plan to change the title of the paper to “Safe In-Context Learning via Risk Control and Early-Exiting”, indicating that our focus is on safety. Additionally, as part of the rebuttal process, **we are currently running a static early-exit model as a baseline for comparison with our dynamic risk-controlled approach, in response to concerns raised by Reviewers dbng and hBNu, and will share results over the next few days**. We are further looking into running an additional experiment on an open-ended generation task, building on an approach such as [1].

We will post results from these experiments over the following days. In the meantime, we respond to your questions and concerns below. **We look forward to addressing any remaining questions during the rebuttal and hope these updates warrant a higher evaluation.**

[1] Tal Schuster, Adam Fisch, Jai Gupta, Mostafa Dehghani, Dara Bahri, Vinh Q. Tran, Yi Tay, Donald Metzler. *Confident Adaptive Language Modeling.* NeurIPS 2022.

---

### Author Response · Authors · 2025-12-03
**Author Final Remarks**

Dear reviewers and ACs, Thank you for your careful consideration of our submission.

## Key Contributions:

- We formulate a *new safe early-exit model* that uses the zero-shot baseline as an explicit reference for safe behavior. To the best of our knowledge, this is the first work to apply dynamic early-exiting for ensuring safety.
- We propose a *novel in context learning (ICL)-specific loss* to measure “overthinking” by quantifying performance relative to the zero-shot model.
- We introduce a *domain-preserving risk transformation* that extends Learn-then-Test (LTT) risk control to losses with negative values, enabling rigorous safety guarantees while still exploiting performance and efficiency gains from helpful demonstrations.
- We perform experiments with a mix of helpful + harmful context on 4 models and 8 tasks (32 settings), repeated with 4 total distributions of in-context demos (Fig. 5, Sec. C). This totals 128 experiments showing that we achieve robust safety guarantees with better performance & efficiency than prior approaches.

## Key Strengths:

We appreciate the reviewers’ positive feedback, which consistently acknowledges the key technical contributions, theoretical rigor, and comprehensive evaluation presented in our work.

**Reviewer dbng:** Our work provides a novel effective framework addressing both risk and efficiency (”effectively controls predictive risk while achieving significant computational speedups”) which is well-supported by theory (”the method is theoretically grounded through the use of a risk control framework”).

**Reviewer hBNu:** Our work provides a rigorous, novel framework for enhancing security in LLMs (”the authors introduce a risk control framework to the LLM security problem [with] sufficient theoretical support”) while simultaneously achieving computational efficiency (”dynamically [deciding] when to early-exit in the reasoning process, thereby avoiding unnecessary computation”).

**Reviewer XUPs:** We effectively balance helpful and harmful demonstrations (“this method can often benefit from correct demonstrations without being harmed much by incorrect demonstrations”), and effectively demonstrate the effectiveness of our approach (”[works] well in many of the studied settings”) through very well-designed settings (”the experiment design seems very thoughtful”).

**Reviewer FF3D:** We present a clear novel problem formulation (”framing ICL safety as a risk control problem relative to a zero-shot baseline is a clear and intuitive formulation”) with significant technical contributions (”the proposed risk transformation…is a solid technical contribution that could be useful for other applications”) supported by thorough evaluations (”evaluation is thorough, covering multiple models, datasets, and proportions of correct/incorrect demonstrations”).

## Key Feedback: Adversarial Risks

We would like to clarify that the security and adversarial risks mentioned in the introduction are intended to build motivation for our general approach, though we do not make a direct claim of contribution in an adversarial setting. Existing adversarial and jailbreak defense techniques often require model re-training; in contrast, our method provides distribution-free safety guarantees without assuming any prior knowledge of specific error types. We do not view these two approaches as direct competitors; rather, they are complementary approaches to LLM safety. We appreciate the reviewers’ feedback on this point and will de-emphasize the adversarial setting in the abstract and introduction.

## Additional Experiments

1. *Dynamic vs static early-exiting.* We justify our design choice of using dynamic early-exiting for risk control, as opposed to static early-exiting (the closest approach to an existing baseline), by repeating our experiments from Fig. 5 using a threshold over a static exit layer instead of model confidence. We find that our approach provides strictly better risk control, as it is much more granular than static early-exit: we can much more accurately approximate the target epsilon risk bound.
2. *A single lambda-value can work well across an aggregation of multiple tasks.* We reproduced Figure 3 from the paper aggregated across all datasets. We find that there still exists a good lambda that loses no more than 5% of the accuracy gains from correct demonstrations while still doing better than the full model given incorrect demonstrations. In general, increasing the granularity of the lambdas (i.e. the number of lambda-values evaluated) can further increase the number of such lambdas. Our approach cannot guarantee the existence of such a lambda, but it is quite a reasonable assumption that one may exist, even when aggregating across many tasks.

We also addressed all comments on language & notation. We trust these revisions demonstrate our commitment to clarity, completeness, and rigor, and hope they will be considered in your final evaluation.

Best regards,

Authors

---

### Meta-Review · Area_Chair_mqwN · 2026-01-06

**Summary:**

The technical contribution is limited and the method was only tested on a single task.

**Reviewer Concerns:**

Some math formulas are revised but the applicability is still questionable.

**Reviewer Scores:**

N/A

---

### Decision · Program_Chairs · 2026-01-26

Reject